# The Correlation Between Abnormal Uterine Artery Flow in the First Trimester and Genetic Thrombophilic Alteration: A Prospective Case-Controlled Pilot Study

**DOI:** 10.3390/diagnostics10090654

**Published:** 2020-08-31

**Authors:** Natalija Vedmedovska, Diana Bokucava, Anda Kivite-Urtane, Vita Rovite, Liene Zake-Nikitina, Janis Klovins, Violeta Fodina, Gilbert G. G. Donders

**Affiliations:** 1Department of Obstetrics and Gynecology, Riga Stradins University, LV-1007 Riga, Latvia; diana.bokuchava@gmail.com; 2Department of Public Health and Epidemiology, Riga Stradins University, LV-1010 Riga, Latvia; anda.kivite-urtane@rsu.lv; 3Latvian Biomedical Research and Study Centre, LV-1067 Riga, Latvia; vita.rovite@biomed.lu.lv (V.R.); liene.nikitina-zake@rsu.lv (L.Z.-N.); klovins@biomed.lu.lv (J.K.); 4Department of Medicine, University of Latvia, LV-1050 Riga, Latvia; 5Reproductive Clinic IVF RIGA, LV-1010 Riga, Latvia; violeta.fodina@ivfriga.eu; 6Department of Obstetrics and Gynecology, University of Antwerp, 2550 Antwerp-Edegem, Belgium; gilbert.donders@gmail.com

**Keywords:** abnormal uterine artery flow, genetic polymorphisms

## Abstract

Introduction. Evaluation of the first trimester uterine artery flow can predict the development of obstetrical complications. A genotype, making women prone to microthrombi. constitutes the main known susceptibility factor for anomalous development of placenta. Our aim was to study whether polymorphisms of 10 genes leading to blood clotting abnormalities are related to abnormal uterine artery blood flow in the first trimester, and may predict placenta-related diseases. Material and methods. In primary analyses we included 19 singleton pregnancies with abnormal blood flow in the uterine arteries during the first trimester of gestation, and 24 matched control with normal flow patterns. All patients were genotyped for sequence variations in F5, F2, F11, MTHFR, SERPINE-1, CYP4V2, SELE, GP6, angiotensinogen (AGT) and fibrinogen gamma (FGG) genes and followed up until delivery. Results. There were no differences between groups regarding selected sequence variations in any of these genes. The co-occurrence of several polymorphisms in the same patient was also not related to the blood flow patterns in the uterine arteries. Conclusions. Although we found certain trends of genetic polymorphisms being related to preeclampsia and fetal growth, we failed to find an association between clotting gene polymorphisms, single or in combination, with the abnormal uterine flow in the first trimester.

## 1. Introduction

Inheritable predisposition to thrombosis increases coagulation in pregnancy and is associated with an elevated risk of miscarriage, intrauterine fetal death, preeclampsia (PE), eclampsia (E) and fetal growth restriction [1]. In 2017, PE/E was diagnosed in 400 (2.0%) and hypertension during pregnancy in 707 (3.4%) of 20,406 births in Latvia, (www.spkc.gov.lv). In the period 2013–2015, maternal mortality due to PE/E was as high as 1.6 per 100,000 live births in Latvia, compared to only 0.08 in the UK (period 2009–2014) [2].

Following Mendelian patterns of disease inheritance, many families have “private” genes predisposing them to micro-thrombi leading to abnormal placental circulation. However, studies of the association of adverse perinatal outcome with factor V Leiden (F5), methylenetetrahydrofolate reductase (MTHFR) and prothrombin (F2) have shown contradictory results [3]. Mutations in *SERPINC1*, the gene encoding antithrombin, are associated with pathogenicity during gestation under specific conditions [4]. The *CYP4V2* gene is known to be involved in blood clotting, but it is unclear how the gene variations may affect placental development [5]. Presence of the polymorphism of the E-selectin gene (*SELE*) predisposes to the development of essential hypertension [6]. A polymorphism of the glycoprotein VI (GP6) gene is also related to the thrombus formation [7]. Still, although polymorphisms in all these genes may contribute to the risk of developing defective placentation in different populations, data are scarce.

Among the factors known to be involved in the pathogenesis of abnormal placentation are inadequate spiral artery modeling and abnormal trophoblast invasion. Doppler ultrasound of the uterine arteries (UtA) enables appraisal of the trophoblast invasion and spiral artery conversion as early as in the first trimester of pregnancy [8]. The effect of aspirin on the resistance of the trophoblastic flow in women with abnormal UtA Doppler in the first trimester in order to prevent hypertensive disorders in pregnancy is not clear [9]. The inconsistency of these results may be explained by the different time frames in which it was given during pregnancies and by the lack of homogeneity of the populations studied [10]. Therefore, we hypothesize that early identification of a high risk profile for developing hypertensive and other disorders related to abnormal placentation can potentially improve the pregnancy outcome through timely pharmacological interventions.

To our knowledge, this study is the first to investigate the link between thrombophilic genetic predispositions and uterine flow characteristics as early as in the first trimester of pregnancy.

Our aim is to study whether genetic polymorphisms causing thrombophilic predisposition are related to abnormal uterine artery blood flow in the first trimester of pregnancy, which reflects abnormal trophoblast invasion.

## 2. Materials and Methods

This prospective case controlled study was performed from January 2015 to December 2017 at Riga Maternity hospital in Latvia, which performs approximately 7000 deliveries per year. Woman with singleton pregnancy attending for their routine first-trimester screening were invited to participate in the study. As funding was lacking for a larger full scaled study at this point, we decided to perform a pilot study to see whether further investment in a larger study would be feasible and worthwhile. The proportion of abnormal uterine artery velocimetry constitutes around 30% of all measurements. According to Sotriadis (2019) et al., bilateral notching is present in 50% of normal pregnancies [11]. Taking into account a prevalence of genetic thrombophilic polymorphisms in the European population from 2–10%, and a three-fold higher prevalence of MTHFR C677T heterozygosity, the required sample size would have to be 377 for prothrombin G20210A/ Factor V Leiden mutations and 63 for MTHFR C677T heterozygosity [12]. Still we felt it was important to study these genetic aberrations in our setting, as we hoped this pilot project would help us to convince the university and potential sponsors to engage in a full study afterwards.

All patients were informed about the goals and methods of the study before enrolment, and signed the Informed Consent form before being included in the study. Both the biobank and prenatal research study have been approved by the Central Medical Ethics Committee of Latvia (protocol No. 22.03.07/A7 and 01.29.1/2 (28.08.2014) respectively.

At 11^+0^ to 13^+6^ weeks of gestation, we recorded maternal characteristics, medical history and ultrasonographic data such as fetal crown-rump length (CRL), nuchal translucency thickness and pulsatility index (PI) of the uterine arteries. Blood samples were taken for maternal serum pregnancy-associated plasma protein-A (PAPP-A), free β-human chorionic gonadotrophin, and genetic analysis. Cases with fetal chromosomal and anatomical anomalies were excluded.

### 2.1. Measurement of Uterine Artery Doppler Flow

Blood flow in both uterine arteries was measured transabdominally at the time of the routine first trimester screening visit, according to study protocol, by one of the two operators (NV, DB) using a 3.5 MHz 3D probe (Voluson E8, GE, Tampa, FL, USA). Abnormal uterine artery velocimetry was considered as the mean PI value above the 95th percentile for gestational age based references ranges [13] and/or bilateral presence of early diastolic «notch». After inclusion of a patient with increased resistance of the uterine arteries, the subsequent patient with normal uterine artery blood flow was asked to serve as a control.

The risk of developing PE and Fetal growth restriction (FGR) as calculated by the Fetal Medicine Foundation (FMF)-2012 software (version 2.81, London, The United Kingdom, The Fetal Medicine Foundation, www.fetalmedicine.org) for every pregnant woman. When the calculated adjusted risk for PE/FGR before 34 weeks was at or above the 10% percentile of the population, it was considered as high risk and, accordingly, a routine protocol 80 mg of acetylsalicylic acid per day was administered.

### 2.2. Detection of Polymorphisms

All study patients and controls were genotyped to detect selected sequence variations in F5, F2, F11 (coagulation factor XI), MTHFR*, SERPINE-1*, *CYP4V2*, *SELE*, *GP6*, *AGT* and *FGG* genes. White blood cell derived DNA was acquired from the Genome Base of the Latvian Population (LGDB) where it was prepared according to a standard phenol-chloroform extraction protocol described there: Design, Goals, and Primary Results [14]. Genotyping was performed using TaqMan SNP Genotyping Assay (ThermoFisher Scientific, Austin, TX, USA) following manufacturer’s instructions on ViiA7 Real-Time system (ThermoFisher Scientific, USA). Genotypes were assigned using AutoCaller 1.1 (ThermoFisher Scientific, USA) software and manually verified.

Information about birth weight, duration of gestation at delivery, mode of delivery, and maternal complications were obtained from standardized medical records. Correct expected delivery dates were ascertained by first trimester dating ultrasonography in all participants.

### 2.3. Statistical Analysis

Statistical Package for the Social Sciences (SPSS) version 20.0 (IBM SPSS Corp., Armonk, NY, USA) was used for statistical analysis. Statistical analysis of the genotype distributions was done using a gene-dosage model (homozygous wild-type vs. heterozygous mutant vs. homozygous mutant). Statistical significance was evaluated at the 0.05 level of significance. Frequencies of categorical variables were compared between groups of patients by using Chi-square or Fisher exact test. The normality of the distribution of parametric variables was checked by using the Kolmogorov-Smirnov test. For variables which did not meet the criteria of normal distribution, Mann-Whitney U test was used to compare variables between Cases and Controls [15,16]. 

## 3. Results

Of a total of 58 patients (29 Cases and 29 Controls), six moved out of town and gave birth elsewhere, where their information could not be retrieved. For another four women, follow-up information was missing at some stage of the study, leaving 48 (83%) remaining to complete the study. Polymorphism studies failed due to technical problems in five women, leaving 43 patients eligible for analysis, consisting of 19 study patients and 24 controls. Altogether 10 different gene polymorphisms were analyzed.

The demographic characteristics of the participating women and dropouts are shown in Table 1. No significant difference between the groups was found for age, BMI and the patient’s place of residence. Both groups were similar in respect to level of education (Table 1), method of conception, gravidity, and parity (Table 2).

Obstetrical characteristics show no differences between the groups regarding most of the complications in current or previous pregnancies (Table 2). Differences in progesterone use during pregnancy (21.1% vs. 8.3%, *p* = 0.38) and the incidence of threatened abortion with severe bleeding (15.8% vs. 4.2%, *p* = 0.31) were not significant between cases and controls, respectively. Study women had no more preeclampsia (8.3% vs. 0, *p* = 0.50) or gestational hypertension (10.5% vs. 4.2%, *p* = 0.58) than controls. Use of aspirin was also similar in both groups (47% vs. 29.2%, *p* = 0.34).

We did not find differences between groups with respect to the rate of stillborn (5.3% vs. 4.2%, *p* = 0.1) or PE (0 vs. 4.2%, *p* = 1) during previous pregnancies, nor were there significant differences regarding extragenital morbidities such as cardiovascular diseases (15.8% vs. 4.2%, *p* = 0. 31) and chronic essential hypertension (0% vs. 12%, *p* = 0.24) in study patients vs. controls, respectively (Table 2).

The first trimester screening shows no differences between the groups regarding embryo size, nuchal translucency, maternal serum PAPP-A and free β-human chorionic gonadotrophin (Table 2)

All 43 analyzed patients including those who had signs of threatened abortion during pregnancy gave birth successfully. Outcome of the current pregnancy showed significant differences: mean gestational age and birth weight at delivery were 39.3 (±2.4 weeks) and 3494 (±586 g) in the study group, vs. 39.8 (±1.1 weeks) and 2988 (±919 g) in the control group, respectively (*p* < 0.04). However, the differences between the groups in rates of newborn gender were not statistically significant, *p* = 0.39 (Table 2).

The polymorphisms of genes are represented in Table 3. There were no differences between both groups regarding the frequency of selected sequence variations in F5, F2, F11, MTHFR*, SERPINE-1*, *CYP4V2*, *SELE*, *GP6*, *AGT* and *FGG* genes. The majority of participants (77.4%) had more than one polymorphism at the same time. However, occurrence of simultaneous polymorphisms of several genes in a patient did not differ between cases and controls (Table 4). After adjustment of polymorphisms for smoking, cardiovascular disease, IVF/ICSI, BMI, and age there were no statisticaly significant associations between selected gene variations and abnormal uterine flow (Table 5).

## 4. Discussion

Inheritable thrombophilia is a risk factor for inadequate uterine-placental circulation. In the present work we studied the hypothesis that genetic polymorphisms can contribute to uterine blood flow changes in order to better understand one of the most mysterious diseases in obstetrics.

The most frequent causes of inherited thrombophilia are the factor V Leiden mutation and the prothrombin gene mutation. According to Nelson et al., (2006) the prevalence of thrombophilia among Europeans varies from 2–10% [12]. The prevalence of Factor V Leiden heterozygosity is 2–7%, MTHFR C677T homozygous is 10% and prothrombin G20210A heterozygosity is 2%. There are no uniform data on the prevalence of these gene polymorphisms in Latvia.

As huge controversies exist regarding the impact of inherited thrombophilia on perinatal outcome, even bigger are the discrepances in opinion about the need for and modalities of screening and prevention [1,17,18,19,20,21]. As a result, a wide variation of preventive actions has been studied, resulting in a predictably great diversity of results [1,22,23,24]. Although the data of our pilot study did not strongly support the use of thrombogenic gene polymorphism testing in the first trimester to predict adverse perinatal outcome, we realized that some trends could have been statistically insignificant if the study sample size bigger. Taking into account the prevalence of genetic thrombophilic polymorphisms, however, the necessary sample size for this study would be at least 377 for prothrombin G20210A heterozygosity and Factor V Leiden heterozygosity and 63 for MTHFR C677T mutations, which was not funded for in this pilot project [12]. A future study focusing on selected gene abnormalities, larger number of cases, more controls per case and closer surveillance of loss to follow-up cases may therefore show more solid results. Indeed, despite bing not statistically different, threatened abortion, hypertensive disorders and history of maternal cardiovascular disease were all three- to four-fold more frequent in our study group than in controls. At the same time two patients from control group developed PE (8.3%), which is higher than the average incidence in Latvia. This is probably due the profile of patients we treat in this referral center, as according to maternity hospital reports 8% of patients were treated with PE in 2016. After this, the diagnostic criteria were sharpened, reducing these figures to 0.4% and 1.7% in 2017 and 2018, respectively, i.e., after the study inclusion period.

We hypothesized that coagulation related gene mutations may disturb invasion of the spiral arteries and cause abnormal uterine artery blood flow, which then could serve as an indication for polymorphism screening early in pregnancy. Stonek et al., (2008), analyzed the correlation between Doppler studies and gene mutation and found that MTHFR C677T T allele at 12 and 22 weeks gestation does not influence uterine artery PI values and «notches» [25]. Previously Druil et al., (2005) failed to find that the frequency of homozygous variations of mutations of MTHFR and prothrombin genes were higher in patients with abnormal uterine artery Doppler flows at 24weeks of pregnancy, but found that carriers of the factor V Leiden mutation had significantly higher abnormal flow in the uterine arteries [26]. Similarly, Bohiltea et al. reported a high mean pulsatility index of the uterine artery at the second and third trimesters in women with factor V Leiden mutation [27]. In our study group, F5 Leiden mutation was not encountered, and both groups had homozygous and heterozygous carriers of *MTHFR* in similar frequencies. None of the other polymorphisms, nor combinations of them in our series were linked to arterial flow abnormalities and impairment of pregnancy outcomes, including hypertensive complications.

Interestingly, no difference in frequency of aspirin use was revealed between groups. Indeed, as aspirin was administered according to risk calculation, taking into account several maternal characteristics, it would have been unethical not to supply aspirin to the controls who qualified for its use. Most likely, other factors than thrombophilia are also responsible for disrupted trophoblastic invasion [28]. The lack of placenta histology also precludes the chance of gaining more insight into the potential effect of multiple polymorphisms on placental architectonics and function.

We have no explanation as to why the birth weight of neonates in the group with increased resistance in the uterine flow during the first trimester was higher than in controls. As there were no significant difference between the groups regarding demographic factors and fetal gender that could influence the newborn’s weight, we can speculate that inhibition of cyclooxygenases due to aspirin use in this group has promoted placental perfusion [29]. The result is in concordance with previous studies [11].

## 5. Conclusions

The data presented in this prospective, controlled study establish that the polymorphisms of F5, F2, F11, *MTHFR*, *SERPINE-1*, *CYP4V2*, *SELE*, GP6, FGG, and AGT genes did not correlate with abnormal uterine artery blood flow in the first trimester compared with control. However, the trend toward more hypertensive disorders and adverse outcome may have been insignificant due to low numbers. There is circumstantial evidence from this and other studies also that early initiation of low dose aspirin may improve placental perfusion and helps to prevent adverse outcome. More and larger studies are necessary, as finding genes predisposing to abnormal placentation would enhance our understanding of the disease mechanisms and would allow identification of prognostic and therapeutic subgroups, in order to modify the dose or explain the mechanism of resistance to it in the different subgroups.

## Figures and Tables

**Table 1 diagnostics-10-00654-t001:** The demographic characteristics of patients from study, controls and drop outs groups.

Variable	Study Group (19)	Control Group (24)	Dropouts (10)	*p* Value
	N of Participant Eligible for Analysis (%)	N of Participants Eligible for Analysis (%)	N of Participants Eligible for Analysis (%)	
**Age (years)**				
≤ 35	13 (68.4)	12 (50.0)	5 (50.0)	0.43
>35	6 (31.6)	12 (50.0)	5 (50.0)	
**Mean (IQR)**	34 (30–36)	33 (29.3–37)		0.72
**BMI**				
<18.5	3 (15.8)	3 (12.5)	3 (30.0)	0.23
18.5–24.9	5 (26.3)	5 (20.8)	3 (30.0)	
25–29.9	9 (47.4)	6 (25.0)	3 (30.0)	
>30	2 (10.5)	10 (41.7)	1 (10.0)	
**Mean (IQR)**	24 (20–27)	24 (19–30)		0.86
**Type of residence**				
**Urban**	17 (89.5%)	24 (100)		0.19
**Rural**	2 (10.5%)	0		
**Level of Education**				
Primary	0	1 (4.2)		0.79
Secondary	3 (15.8)	2 (8.3)		
Secondary professional	4 (21.1)	4 (16.7)		
High/university	12 (63.2)	17 (70.8)		

Data are given as numbers, percentages, and the total for that group in parentheses.

**Table 2 diagnostics-10-00654-t002:** Obstetrical characteristics and concurrent medical problems of patients from study and control groups.

Variable			Study Group (19)		Control Group (24)		*p* Value
			N	%	*n*	%	
Stillbirth in previous pregnancies			1	5.3	1	4.2	1.0
**Gravidity**							
Mean (IQR)			1 (1–3)		2 (1–3)		0.33
**Parity**							
Mean (IQR)			1 (1–1)		1 (1–2)		0.21
**Method of conceiving**							
spontaneous			12	63.2	17	70.8	0.73
IVF/ICSI			6	31.6	7	29.2	
insemination			1	5.3	0	0	
Threatened abortion			3	15.8	1	4.2	0.31
Progesterone use (p/o; t/v)			4	21.1	2	8.3	0.38
Gestational hypertension			2	10.5	1	4.2	0.58
Preeclampsia			0	0	2	8.3	0.50
Aspirin use 80mg/24h			9	47.4	7	29.2	0.34
Concurrent medical problems							
Kidney and urinary disorders			0	0	2	8.3	0.50
Cardiovascular diseases			3	15.8	1	4.2	0.31
Smoking			3	15.8	0	0	0.08
**CRL (mm)**							
Median (IQR)			66 (63.5–76)		66 (62.3–72.8)		0.45
**NT**							
Median (IQR)			1.8 (1.4–2.3)		1.8 (1.5–1.9)		0.99
**PAPP-A (MOM)**							
Median (IQR)			1.2 (0.6–1.5)		1.0 (0.7–1.9)		0.84
**Beta-hCG (MOM)**							
Median (IQR)			1.1 (0.9–1.6)		1.3 (0.9–2.0)		0.69
**Weight of Newborn (g)**							
Mean (SD), median	3494.00 (586.89)	3565	2988.13 (919.29)		3235		0.04
**Gender**							
Male	12	63	12		50		0.39
Female	7	36.8	12		50		
**Delivery (weeks)**							
Mean (SD), median	39.31 (2.42)	39.79	39.83 (1.18)		40.14		0.71

**Table 3 diagnostics-10-00654-t003:** Polymorphisms in the study and control groups according to type.

Genetic Variations	Study Group		Control Group		*p* Value
	N	%	*n*	%	
**rs6025- F5 Association-VT/PE**					
C_C (wild type)	19	100.0	23	95.8	1.0
C_T	0	0	1	4.2	
**rs1799963 -F2 Association -VT/PE**					
G_G (wild type)	18	94.7	24	100.0	0.44
A_G	1	5.3	0	0	
**rs699-AGT Association -PE**					
A_A	3	15.8	7	29.2	0.52
A_G	9	47.4	11	45.8	
G_G	7	36.8	6	25.0	
**rs1801133-MTHFR Association -PE**					
G_G (wild type)	4	21.1	10	41.7	0.43
G_A	10	52.6	8	33.3	
A_A	3	15.8	5	20.8	
G_T	1	5.3	1	4.2	
T_T	1	5.3	0	0	
**rs2066865-FGG Association -VT**					
G_G (wild type)	13	68.4	17	70.8	0.92
A_G	5	26.3	6	25.0	
G_A	1	5.3	0	0	
A_A	0	0	1	4.2	
**rs2227589- SERPINC1 Association-VT**					
C_C (wild type)	16	84.2	19	79.2	0.42
C_T	2	10.5	5	20.8	
T_T	1	5.3	0	0	
**rs1613662-GP6 Association-VT**					
A_A (wild type)	12	63.2	21	87.5	0.07
A_G	4	21.1	3	12.5	
G_G	3	15.8	0	0	
**rs13146272- CYP4V2 Association-VT**					
A_A (wild type)	7	36.8	12	50.0	0.75
A_C	10	52.6	9	37.5	
C_C	2	10.5	3	12.5	
**rs2289252-F11 Association-VT**					
C_C (wild type)	10	52.6	6	25.0	0.20
C_T	8	42.1	15	62.5	
T_T	1	5.3	3	12.5	
**rs5361-SELE Association-VT**					
T_T (wild type)	17	89.5	16	66.7	0.08
G_T	2	10.5	8	33.3	

PE-preeclampsia, VT-venous thrombosis.

**Table 4 diagnostics-10-00654-t004:** Number of polymorphisms in both groups.

No of Polymorphisms	Study Group		Control Group		*p* Value
	*n*	%	*n*	%	0.37
1	1	5.3	1	4.2	
2	1	5.3	6	25.0	
3	4	21.1	4	16.7	
4+	13	68.4	13	54.2	
Total	19	100.0	24	100.0	

**Table 5 diagnostics-10-00654-t005:** Association between polymorphisms and abnormal a.uterina flow in univariate and multivariate analysis.

Genetic Variations	OR	95% CI	*p*	aOR *	95% CI	*p*
**rs6025-F5 Association-VT/PE**						
C_C (wild type)	1			1		
C_T	NA	NA	NA	NA	NA	NA
**rs1799963 -F2 Association-VT/PE**						
G_G (wild type)	1			1		
A_G	NA	NA	NA	NA	NA	NA
**rs699-AGT Association-PE**						
A_A	1			1		
A_G	1.9	0.4–9.6	0.26	2.7	0.3–28.2	0.41
G_G	2.7	0.5–15.5	0.43	1.9	0.2–24.9	0.61
**rs1801133-MTHFR Association-PE**						
G_G (wild type)	1			1		
G_A	3.1	0.7–13.8	0.13	3.5	0.3–38.3	0.30
A_A	1.5	0.2–9.5	0.67	1.8	0.1–32.9	0.71
G_T	2.5	0.1–50.4	0.55	5.4	0.1–278.8	0.40
T_T	NA	NA	NA	NA	NA	NA
**rs2066865-FGG Association-VT**						
G_G (wild type)	1			1		
A_G	1.1	0.3–4.4	0.9	1.1	0.2–6.7	0.89
G_A	NA	NA	NA	NA	NA	NA
A_A	NA	NA	NA	NA	NA	NA
**rs2227589-SERPINC1 Association-VT**						
C_C (wild type)	1			1		
C_T	0.5	0.1–2.8	0.41	0.4	0.03–6.1	0.54
T_T	NA	NA	NA	NA	NA	NA
**rs1613662-GP6 Association-VT**						
A_A (wild type)	1			1		
A_G	2.3	0.4–12.2	0.32	1.9	0.3–14.2	0.52
G_G	NA	NA	NA	NA	NA	NA
**rs13146272-CYP4V2 Association-VT**						
A_A (wild type)	1			1		
A_C	1.9	0.5–7.0	0.33	3.2	0.5–20.1	0.22
C_C	1.1	0.2–8.6	0.90	3.5	0.2–53.5	0.37
**rs2289252-F11 Association-VT**						
C_C (wild type)	1			1		
C_T	0.32	0.1–1.2	0.09	0.2	0.02–1.4	0.10
T_T	0.2	0.02–2.4	0.20	0.1	0.004–4.2	0.24
**rs5361-SELE Association-VT**						
T_T (wild type)	1			1		
G_T	0.2	0.04–1.3	0.09	0.1	0.01–1.9	0.13

* aOR—adjusted odds ratio, adjusted for harmful health behaviour, cardiovascular disease, IVF/ICSI, BMI, age.

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
