# Peer review of "The Correlation Between Abnormal Uterine Artery Flow in the First Trimester and Genetic Thrombophilic Alteration: A Prospective Case-Controlled Pilot Study"

_diagnostics, 2020, doi:10.3390/diagnostics10090654_

Round 1

Reviewer 1 Report

This set of data are of interest in analyzing the correlation between thrombophilic related gene polymorphisms and abnormal uterine artery flow in the first trimester. This prospective study included 19 cases with abnormal uterine artery flow during the first trimester of gestation and 24 cases with normal flow. Several thrombophilic related gene polymorphisms were detected between the two groups, although the authors failed to identify the association between gene polymorphisms and abnormal uterine artery flow in the first trimester. These results are of interest. However, there are some major concerns regarding to study design, data analysis, result interpretation need to be addressed.  

  1. The major concern for this study is the sample size and demographic characteristics between the study and control groups. The authors need to perform sample size calculation to confirm the minimum sample size for this case control study. This reviewer understand this is a prospective study that requires the consent from the participants, which may be the reason for small sample size. However, this study was performed in a hospital with 7000 deliveries per year within three years from January 2015 to December 2017, which means approximately 21000 deliveries in this time frame. What are the reasons for low inclusion rate? What’s the overall incidence of abnormal uterine artery velocimetry when screening for the participants in the study group? The authors need to provide more detailed information regarding to the inclusion and exclusion processes in this study.
  2. In Abstract, the authors stated that 29 cases in study group and 29 cases in control group were included in this study. however, only 19 cases in study group and 24 cases were included in the analysis of primary outcomes. This information needs to be revised.
  3. In study group, there are 3 cases with cardiovascular diseases, and 3 cases with smoking history, and the p value regarding to smoking is 0.08 between the two groups. It is possible that these conditions may cause uterine artery flow abnormalities. Theoretically, these cases should be excluded in this study. Again, the authors need to provide more information about the inclusion and exclusion criteria, the only exclusion criteria in the Material and Methods is cases with fetal chromosomal and anatomical anomalies were excluded.
  4. The authors stated that the incidence of PE/E is 2.0 % in Latvia in 2017. However, in control group, the incidence of PE reaches to 8.3%, which is much higher than the average incidence in Latvia. What’s the incidence of PE in this hospital during the study time frame?
  5. The authors indicate that the blood samples have been taken for serum PAPP-A and beta-hCG, nuchal translucency thickness and fetal crown-rump length were detected during the first trimester, is there any differences regarding these parameters between the two groups?
  6. Table 1, the authors need to provide data analysis for average age, average BMI, gravidity and parity between the two groups.
  7. Please indicate the definition of AUt in Table 2. Is it the abnormal uterine artery flow? If so, it is interesting that the cases with AUt (> 95th centile) at the second and third trimesters reaches to 75% and 81.8% in the study and control groups, respectively. What is the average incidence of AUt (> 95th centile) in this obstetric center? Line 85 to 87, reference 11 (Reference ranges for uterine artery mean pulsatility index at 11-41 weeks of gestation) is based on the population from Barcelona, Spain, is there any difference in the reference ranges between Spanish and Latvia?
  8. In Table 2, 3 cases in study group, and 1 cases in control group had threatened abortion, what’s the outcome of pregnancy in these 4 cases. Did the 43 cases included in the final analysis all gave birth?
  9. Table 2, the weight of newborn is significantly increased in the study group compared to control group (3594 vs. 2988), while the average gestational age is similar between the two groups. The authors explain that the use of aspirin may increase the placental infusion and contribute to the increased body weight, however, the incidence of aspirin use is similar between the two groups. The authors still need to analyze the demographic characteristics between the two groups. For example, what is the sex ratio of newborns between the two groups?
  10. Although the pathogenesis of preeclampsia and venous thrombosis and other pregnant complications remain unclear, it is accepted that these complications are caused by multiple factors, which means they might not be triggered by single factor or single gene. The authors should perform the Logistic regression analysis to analyze the correlation of gene polymorphisms and abnormal uterine artery flow.
  11. Minor errors:

Line 60: “dorders” should be “disorders”

Line 141: “vs” should be “vs.”

Line 143: 3.494 (+ 586 g) should be 3,494 or 3494; 2.988 should be 2,988 or 2988.

Table 2. > 95thcentile should be 95th centile.

Table 2. “PE/Gestationa hypertension” should be “Gestational”, and the format is wrong in this row.

Author Response

Response to the reviewer Nr 1.

  1. The major concern for this study is the sample size and demographic characteristics between the study and control groups. The authors need to perform sample size calculation to confirm the minimum sample size for this case control study. This reviewer understand this is a prospective study that requires the consent from the participants, which may be the reason for small sample size. However, this study was performed in a hospital with 7000 deliveries per year within three years from January 2015 to December 2017, which means approximately 21000 deliveries in this time frame. What are the reasons for low inclusion rate? What’s the overall incidence of abnormal uterine artery velocimetry when screening for the participants in the study group? The authors need to provide more detailed information regarding to the inclusion and exclusion processes in this study.

Thank you for this remark. It bothers us also that we could not perform a bigger study. First of all, being a referral hospital, the majority of the patients giving birth at our hospital have not had a 1st trimester screening in our unit. Second, most importantly, funding was lacking for a full-scale study. For that reason, we were only entitled to perform a pilot study, to see whether further investment in a larger study would be feasible and worthwhile. To our opinion, in many projects too much money and resources are spent in studies that are not appropriate, wasting valuable investments in time and expenses. Indeed the sample size needed for reaching statistical differences in a full-scale study would need to be a 10-fold of what we were able to do. The proportion of abnormal uterine artery velocimetry constitutes around 30% of all measurments. According  Sotriadis (2019), bilateral notching is present in 50% of normal pregnancies. Taking into account a prevalence of genetic thrombophilic polymorphisms in European population from 2 to 10%,  and a 3 fold higher prevalence of MTHFR C677T heterozygicity,  the required sample size should be at 377 for  prothrombin G20210A/ Factor V Leiden and 63 for MTHFR C677T heterozygosity (Nelson, 2006). Still we felt it was important to study these genetic aberrations in our setting, as we hoped this pilot project would help us to convince the university and potential sponsors to engage in a full study afterwards. We introduced this reasoning in the text  (lines 77-87).

Expected frequency of polymorphisms in controls (%)

Expected frequency of polymorphisms in cases (%)

Sample size, cases

Sample size, controls

2

4

1146

1146

2

6

377

377

2

8

208

208

2

10

139

139

10

20

201

201

10

30

63

63

10

40

33

33

10

50

21

21

  1. In Abstract, the authors stated that 29 cases in study group and 29 cases in control group were included in this study. however, only 19 cases in study group and 24 cases were included in the analysis of primary outcomes. This information needs to be revised.

We revised the statement as recommended by reviewer: «In primary analyses we included 19 singleton pregnancies with abnormal uterine arteries blood flow during the 1st trimester of gestation, and 24 matched control with normal flow patterns». (lines 27-29).

  1. In study group, there are 3 cases with cardiovascular diseases, and 3 cases with smoking history, and the p value regarding to smoking is 0.08 between the two groups. It is possible that these conditions may cause uterine artery flow abnormalities. Theoretically, these cases should be excluded in this study. Again, the authors need to provide more information about the inclusion and exclusion criteria, the only exclusion criteria in the Material and Methods is cases with fetal chromosomal and anatomical anomalies were excluded.

We left 3 patients as cardiovascular disorders were neirocirculatory asthenia, which is not considered a condition with real cardiovascular hemodynamic consequences. We decided not to remove the smokers from the analyses as logistic regression was performed. Also, all 3 smokers reported smoking 1 to 6 cigaretes per day only. After adjustment for smoking, cardiovascular disease, IVF/ICSI, BMI, and age there were no statisticaly significant associations between selected gene variations and abnormal uterine flow (Table 5), lines 177-179; 189-193.

  1. The authors stated that the incidence of PE/E is 2.0 % in Latvia in 2017. However, in control group, the incidence of PE reaches to 8.3%, which is much higher than the average incidence in Latvia. What’s the incidence of PE in this hospital during the study time frame?

Thank you for this remark, which is well taken. The high percentage is related to the type of patients we treat as a referral hospital, as according to Matrenity hospital report 8% of patients were treated with PE in 2016. After this, the diagnostic criteria were sharpened, reducing these figures 0.4% and 1.7% in 2017 and 2018, respectively (after the study inclusion). We have included this information in the discussion part (lines 218-222)

  1. The authors indicate that the blood samples have been taken for serum PAPP-A and beta-hCG, nuchal translucency thickness and fetal crown-rump length were detected during the first trimester, is there any differences regarding these parameters between the two groups?

Agree, thank you! The data are providided in the table 2 and discussed in the text-lines 163-165.

  1. Table 1, the authors need to provide data analysis for average age, average BMI, gravidity and parity between the two groups.

The data are included in the table 1.

  1. Please indicate the definition of AUt in Table 2. Is it the abnormal uterine artery flow? If so, it is interesting that the cases with AUt (> 95thcentile) at the second and third trimesters reaches to 75% and 81.8% in the study and control groups, respectively. What is the average incidence of AUt (> 95th centile) in this obstetric center? Line 85 to 87, reference 11 (Reference ranges for uterine artery mean pulsatility index at 11-41 weeks of gestation) is based on the population from Barcelona, Spain, is there any difference in the reference ranges between Spanish and Latvia?

At the second and 3rd trimester the mean PI of both a.uterina >95%  was considered as abnormal. In Latvia we use the reference ranges of Barcelona group, as we do not have our own population based references. We removed the data from the table and discussion part as some missing data on this values in all patients did not  allow to performe correct analyses. Thank you for this comment.

  1. In Table 2, 3 cases in study group, and 1 cases in control group had threatened abortion, what’s the outcome of pregnancy in these 4 cases. Did the 43 cases included in the final analysis all gave birth?

All analysed patients gave birth. We included this information in the discussion part (lines 166-167).

  1. Table 2, the weight of newborn is significantly increased in the study group compared to control group (3594 vs. 2988), while the average gestational age is similar between the two groups. The authors explain that the use of aspirin may increase the placental infusion and contribute to the increased body weight, however, the incidence of aspirin use is similar between the two groups. The authors still need to analyze the demographic characteristics between the two groups. For example, what is the sex ratio of newborns between the two groups?

This point is well taken, and it struck us also to see this unexpected birth weight difference between groups. The different demographic phactors (place of living, education) as well as gender of fetuses were compared and represented in the tables 1 and 2, as well as described in the results and discussion parts:

No significant difference between the groups was found for age, BMI and the patient’s place of residence. Both groups were similar in respect to level of education (Table 1), method of conception, gravidity and parity (Table 2). (Line 139-141). Also the differences between the groups in rates of newborns gender were not statistically significant, p=0.39 (Table 2), line 170-171.

As there were no  significant difference between the groups regarding demographic factors and fetal gender that could influence the newborn’s weight, we can speculate that inhibition of cyclooxygenases due to aspirin use in this group has promoted placental perfusion, especially in the group with predestined hypertensive disorders  (line 245-248)

  1. Although the pathogenesis of preeclampsia and venous thrombosis and other pregnant complications remain unclear, it is accepted that these complications are caused by multiple factors, which means they might not be triggered by single factor or single gene. The authors should perform the Logistic regression analysis to analyze the correlation of gene polymorphisms and abnormal uterine artery flow.

The regression analyses was performed. The table Nr 5 represents the results. After adjustment of polymorphisms for smoking, cardiovascular disease, IVF/ICSI, BMI, and age there were no statisticaly significant associations between selected gene variations and abnormal uterine flow (Line 177-179).

  1. Minor errors:

Line 60: “dorders” should be “disorders”

Line 141: “vs” should be “vs.”

Line 143: 3.494 (+ 586 g) should be 3,494 or 3494; 2.988 should be 2,988 or 2988.

Table 2. > 95thcentile should be 95th centile.

Table 2. “PE/Gestationa hypertension” should be “Gestational”, and the format is wrong in this row.

All  mistakes were corrected

Reviewer 2 Report

The manuscript has a very important subject concerning to obstetrics practice. But, there are several methodological gaps that compromise the correct evaluation of the manuscript data, as follow:

  1. The manuscript assessed gene polymorphism in a very small population. We need to believe that those results did not occured by chance. So, the author have to show the usual frequency of the evaluated genetic alteration in the general population. They need to remember that polymorphism occur in 1% of the population and you need at least 100 patients by group to considere a real polymorphism occurence. An alternative method to access genetic alterations in a small controlled group could be next generation sequencing. 
  2. I suggest to change the term "polymorphism"  for genetic alterations serch in a,b,c,d,e f, genes...in the manuscript title. Even as a pilot study, the author should have calculated the required number of patients by group using SNPs techonologies. Real time for SNPs needs a huge number os individuals. 
  3. Please, support the selected number of patients by group using a statistical model or support the allelic frequency  of those gene polymorphism usinf the literature.

Author Response

Response to the reviewer Nr 2.

  1. The manuscript assessed gene polymorphism in a very small population. We need to believe that those results did not occured by chance. So, the author have to show the usual frequency of the evaluated genetic alteration in the general population. They need to remember that polymorphism occur in 1% of the population and you need at least 100 patients by group to considere a real polymorphism occurence. An alternative method to access genetic alterations in a small controlled group could be next generation sequencing. 

We calculated the necessary amount of patients. Taking into account the prevalence of genetic thrombophilic polymorphisms in European population varies from 2 to 10%, therefore the necessary sample size should be at least 377 for  prothrombin G20210A heterozygosity and Factor V Leiden heterozygosity and 63 for MTHFR C677T, which prevalence in the population reported to be higher (Nelson, 2006). We agree that next generation sequencing could be the answer, but this would require even more funding…

  1. I suggest to change the term "polymorphism"  for genetic alterations serch in a,b,c,d,e f, genes...in the manuscript title. Even as a pilot study, the author should have calculated the required number of patients by group using SNPs techonologies. Real time for SNPs needs a huge number os individuals. 

The title was changed on “The correlation between abnormal uterine artery flow in the first trimester and genetic thrombophilic alteration: a prospective case-controlled pilot study”. In the future studies we will certainly study the possibility to use real time SNP technology. Thank you for the advice!

  1. Please, support the selected number of patients by group using a statistical model or support the allelic frequency  of those gene polymorphism usinf the literature.

As in this pilot study the numbers were too low for splitting the data into frequency groups with specific alleles, we lumped the polymorphisms in order to get stronger data. But we agree that in future studies – if sponsorship gets obtained - we should use a statistical model based on literature allelic frequencies. Thank you again for the wise advice.

We insist in thanking both reviewers for the extensive work that was done and for the major improvement brought into the manuscript. We really appreciated this a lot! In the future work we would be happy to collaborate with you, in case afterwards you would be willing to disclose your contact details. 

Round 2

Reviewer 1 Report

All the points were addressed in the revision.

Reviewer 2 Report

The authors modified important pieces of their work. I insiste that this manuscript has a low number of sample to consider the results as representative, but as a pilot study we wish that, in a soon future, the authors could confirm the significance of their data.